

# Influences of ABO blood group, age and gender on plasma coagulation factor VIII, fibrinogen, von Willebrand factor and ADAMTS13 levels in a Chinese population

Zongkui Wang, Miaomiao Dou, Xi Du, Li Ma, Pan Sun, Haijun Cao, Shengliang Ye, Peng Jiang, Fengjuan Liu, Fangzhao Lin, Rong Zhang and Changqing Li

Institute of Blood Transfusion, Chinese Academy of Medical Sciences & Peking Union Medical College, Chengdu, Sichuan, China

Corresponding authors
Rong Zhang, 444306493@qq.com
Changqing Li, lichangqing268@163.com

## ABSTRACT

**Background.** ABO blood group is a hereditary factor of plasma levels of coagulation factor VIII (FVIII) and von Willebrand factor (VWF). Age and gender have been shown to influence FVIII, VWF, fibrinogen (Fbg), and ADAMTS13 (A disintegrin and metalloprotease with thrombospondin type 1 motif, 13). We investigated the effects of ABO type, age, and gender on plasma levels of FVIII, Fbg, VWF, and ADAMTS13 in a Chinese population.

**Methods.** A total of 290 healthy volunteers were eligible for this study. ABO blood group was determined by indirect technique. FVIII:C and Fbg were measured by clotting assays. VWF antigen (VWF:Ag), collagen-binding activity (VWF:CBA), and ADAMTS13 antigen were assessed by ELISA, whereas VWF ristocetin cofactor activity (VWF:Rcof) was performed by agglutination of platelets with ristocetin.

**Results.** Mean FVIII:C and VWF levels (VWF:Ag, VWF:CBA, and VWF:Rcof) were significantly higher in non-O than in O type subjects ($p < 0.05$ for all comparison). ADAMTS13 antigen decreased with increasing age, whereas the other parameters increased. Other than ADAMTS13 ($p < 0.01$), no gender-related variations were observed in the other parameters. Moreover, FVIII:C, Fbg, VWF:Ag, VWF:CBA, and VWF:Rcof showed significant and positive relationships with age ($r = 0.421, 0.445, 0.410, 0.401,$ and $0.589$, resp.; all $p < 0.001$), whereas a negative relationship was observed for ADAMTS13 antigen ($r = 0.306; p = 0.006$). Furthermore, FVIII:C were strongly correlated with VWF:Ag, VWF:CBA, and VWF:Rcof ($r = 0.746, r = 0.746,$ and $r = 0.576$, resp.; $p < 0.0001$). VWF parameters were also strongly correlated with each other ($r = 0.0847$ for VWF:Ag and VWF:CBA; $r = 0.722$ for VWF:Ag and VWF:Rcof; $p < 0.0001$).

**Conclusions.** ABO blood group, age, and gender showed different effects on plasma levels of FVIII:C, Fbg, VWF:Ag, VWF:CBA, VWF:Rcof, and ADAMTS13 antigen. These new data on a Chinese population are quite helpful to compare with other ethnic groups.

## INTRODUCTION

The ABO blood group, first recognized by Landsteiner in 1900, is the most clinically important of the blood group systems. The antigens of the ABO(H) system (A, B, and H determinants) consist of A carbohydrate structure and/or B carbohydrate structure on the substrate H core structure (*Moll et al., 2014*). Although these structures are traditionally regarded as red cell antigens, A and B carbohydrate structures are actually expressed on a variety of human tissues, including epithelium, sensory neurons, platelets and vascular endothelium (*Eastlund, 1998*).

An association between ABO blood group and plasma Von Willebrand factor (VWF) and coagulation factor VIII (FVIII) levels has been recognized. In both arterial and venous thromboembolism, non-O (A, B, or AB) individuals show significantly increased risk (*Franchini & Mannucci, 2014*; *Liumbruno & Franchini, 2013*), whereas group O individuals have more pronounced inherited bleeding tendency and von Willebrand disease (*Gallinaro et al., 2008*; *Rios et al., 2012*). A scientific basis for this result is that ABO blood group determines the two coagulation glycoproteins levels VWF and FVIII (*Song et al., 2015*). A large twin study (*Orstavik et al., 1985*) has demonstrated that 30% of plasma VWF levels depend on the effect of ABO group. Plasma VWF levels are approximately 25%–30% lower in group O subjects than in non-O individuals (*Franchini & Mannucci, 2014*; *Liumbruno & Franchini, 2013*; *O'Donnell et al., 2005*; *Orstavik et al., 1985*; *Sousa et al., 2007*). Although the mechanisms between ABO blood group and VWF levels are not fully understood, the effects are mediated by the N-linked oligosaccharide chains of VWF, which are similar to the ABO antigens (carbohydrate structure) (*Gallinaro et al., 2008*; *Lynch & Lane, 2016*; *McKinnon et al., 2008*). Alternatively, H antigen expression could mediate the ABO effect on plasma VWF level. Moreover, some studies indicate that the proteolysis of VWF by ADAMTS13 is also affected by ABO antigens. The proteolysis appears to be faster in group O subjects than in non-O carriers (*O'Donnell et al., 2005*). The differences in glycane structure between group O and non-O individuals could alter the susceptibility of VWF for cleavage by ADAMTS13 (a disintegrin-like and metalloprotease with thrombospondin type I motif, 13), which affects the conformation of VWF and the accessibility of the ADAMTS13 cleavage site (*Casari et al., 2013*; *O'Donnell et al., 2005*; *Rios et al., 2012*). The effects of ABO blood groups on FVIII are mainly mediated by an effect on VWF antigen levels, but several other factors (e.g., pregnancy and use of oral contraceptive) are also correlated with FVIII levels (*Rios et al., 2012*; *Song et al., 2015*).

Multiple reports have demonstrated the associations of VWF and FVIII with race, gender, and age. Mean levels of VWF and FVIII are significantly higher in females than in males and in African-American than in Caucasians (*Payne et al., 2014*; *Zhou et al., 2014*). Linear increase with increasing age is observed for VWF, FVIII, and fibrinogen (Fbg) (*Cohen et al., 2012*; *Cowman et al., 2015*; *Ishikawa et al., 1997*). ADAMTS13 shows negative linear association with age and is higher in females than in males (*Mannucci et al., 2001*).

But despite numerous studies that are generally reported on other Asian populations, Caucasians, and Africans, little is known about the Chinese population. With the development of science, it has been gradually realized that there are many ethnicity specific

variations in physiological processes (*Song et al., 2015*). Consequently, the present study was conducted to investigate the influences of ABO blood group, age, and gender on plasma levels of FVIII, Fbg, VWF, and ADAMTS13 in a Chinese population.

## MATERIALS AND METHODS

### Ethics statement

In this study, the recruited volunteers provided anonymous informed consent, with the information of age, and gender, according to the requirements of the Ethics Committee of the Institute of Blood Transfusion. All procedures concerning the experiments with human plasma had been given prior approval by the Department of Public Health of Sichuan Province and the Academic Board of Institute of Blood Transfusion. All clinical research was conducted according to the principles revealed in the Declaration of Helsinki.

### Subjects

In all, 290 healthy volunteers were enrolled from normal blood donors aged 22 y–56 y at Guanghan Plasmapheresis Center and Changning Plasmapheresis Center, which are located in geographically diverse regions of Sichuan province. Inclusion criteria were that all volunteers were ≥18 years, healthy and unrelated. Individuals who had prior history of thrombus or hemorrhage, usage of oral anticoagulation therapy, hepatic disease, HIV infection, pregnancy, diabetes, renal insufficiency and others were excluded from this study by standardized questionnaire according to the "Administrative Measures for Plasmapheresis Center". Plasma samples were drawn in 0.129 M sodium citrate in 9:1 volume ratio by plasmapheresis. Then, aliquots were obtained and stored at $-70\,^{\circ}$C until analysis. Forty-one samples were excluded from further analysis because of clot formation during thawing, and therefore, 249 samples were used for the influences of ABO blood group on plasma FVIII, Fbg, VWF and ADAMTS13 levels. Of the 249 samples, only 121 samples had information on age and gender (male, $n = 49$; female, $n = 72$), which could be used for the further analysis of the influences of age and gender on FVIII, Fbg, VWF and ADAMTS13. The samples were arbitrarily divided into the three following age categories: <40 y (22 y–39 y, $n = 25$), 40 y–49 y ($n = 73$), and >50 (50 y–56 y, $n = 23$) (Table 1).

### Laboratory assays

All samples were tested in triplicate. The ABO blood group phenotype was determined by routine serological test with monoclonal anti-A and anti-B antibodies (STAC Medical Science & Technology Co., Ltd., Jinan, China). FVIII coagulation activity (FVIII:C) and Fbg were measured by clotting assays using commercially available kits (Chengdu Union Biotechnology Co., Ltd., Chengdu, China) on a CA-1500 automated coagulation analyzer (Sysmex Corporation, Kobe, Japan).

VWF antigen (VWF:Ag) was measured by enzyme-linked immunosorbent assay (ELISA). Ninety-six-well plates were coated with rabbit polyclonal anti-human VWF antibodies (DakoCytomation, Glostrup, Denmark) at 4 °C overnight. And then, the plates were blocked for nonspecific binding with PBS containing 2% bovine serum albumin. After incubation at 37 °C for 2 h, the plates were washed four times, plasma samples were added,

**Table 1** Demographics of this cohort.

| Group | Number of subjects[a] |
|---|---|
| **Blood group** | **249** |
| Non-O group | 166 |
| A | 81 |
| B | 69 |
| AB | 16 |
| O group | 83 |
| **Gender** | **121** |
| Female | 72 |
| Male | 49 |
| **Age (year)** | **121** |
| <40 | 25 |
| 40–49 | 73 |
| >50 | 23 |

**Notes.**

[a] A total of 249 donors were included in the final analyses for the effect of ABO Blood Group; Only 121 subjects provided the information of age and gender, which could be used for the analysis of age and gender.

and the plates were incubated for another 2 h at 37 °C. Then, the microwells were incubated with a rabbit polyclonal anti-human VWF conjugated with horseradish peroxidase (Dako-Cytomation) diluted to 1:6000 (217 $\mu$g/L, pH 7.4) in PBS/Tween 20 at 37 °C for 1 h. After further washing, the peroxidase substrate tetramethylbenzidine (TMB) was introduced. The reaction was stopped with 2 M $H_2SO_4$ and the optical density was measured at 450 nm using a spectrophotometer (Molecular Devices, Sunnyvale, CA, USA).

VWF collagen-binding activity (VWF:CBA) was assessed by ELISA similar to VWF:Ag. After overnight coating with type III collagen (2.5 $\mu$g/mL) at 4 °C, microwell plates were incubated with sample at 37 °C for 2 h. Then, a rabbit polyclonal anti-human VWF conjugated with horseradish peroxidase (DakoCytomation) diluted to 1:6000 (217 $\mu$g/L, pH 7.4) was added. The mixtures were incubated at 37 °C for 1 h. After introducing TMB, the optical density was measured.

VWF ristocetin cofactor activity (VWF:Rcof) was measured by agglutination of lyophilized platelets (HYPHEN BioMed, Neuville sur Oise, France) reconstituted with 0.05 M TBS (pH 7.35) and ristocetin at a final concentration of 1.25 mg/mL.

For FVIII:C, Fbg, VWF:Ag, VWF:CBA, and VWF:Rcof, a commercial plasma calibrator (NIBSC, Hertfordshire, UK) was used as the reference plasma. Dilutions of 100% reference plasma were used to construct standard curves for calibration.

ADAMTS13 antigen was tested using a commercial ELISA kit (Sekisui Diagnostics, LLC, CT, USA) in accordance with the manufacturer's recommendations.

## Statistical analysis

Kolmogorov–Smirnov test was used for the normal distribution of all data. Values were expressed as means and standard deviation (SD) or as medians and 25th–75th quartile when appropriate. Multi-group comparisons (different ABO groups and age categories) were conducted by one-way ANOVA followed by LSD post hoc test. The effect of gender on

FVIII, VWF, ADAMTS13, and Fbg was accomplished using two-tailed unpaired Student's *t*-tests. Poisson analysis was used to determine the relationship between age and plasma levels of FVIII, Fbg, VWF, and ADAMTS13. Bivariate correlation analysis was used to calculate the associations of FVIII:C and VWF:Ag, VWF:CBA and VWF:Ag, and VWF:Rcof and VWF:Ag. A correlation coefficient ($r$) of 0.10–0.29 indicates a small correlation, 0.30–0.49 a moderate correlation and 0.50–1.0 a high correlation. A 95% CI [2.5%–97.5%] was used and a *p*-value <0.05 was considered statistically significant. Statistical analyses were conducted using SPSS statistics software version 17.0 (SPSS Inc., Chicago, USA).

## RESULTS

### Effect of ABO blood type, age and gender on FVIII:C

The well-known differences in FVIII:C between ABO blood types were observed, with significantly higher levels in non-O than O individuals ($101.3 \pm 26.4\%$ vs. $74.8\% \pm 23.2\%$, $p < 0.001$), whereas no significant differences were noted among non-O group members (A, AB, and B). FVIII:C showed significant increases with age, with an overall change of 1.41-fold by >50y ($78.2 \pm 15.4\%$ vs. $94.1 \pm 25.0\%$ vs. $110.3 \pm 26.3\%$, $p < 0.001$) (Fig. 1A). Consequently, a moderate and significant correlation ($r = 0.421$, $p = 0.0001$) between FVIII:C levels and age was found (Fig. 2A). Furthermore, no gender-related discrepancies were observed for FVIII:C (Fig. 1A).

### Effect of ABO blood type, age and gender on Fbg

ABO blood type showed no effect on Fbg level, and similarly between genders. However, Fbg levels increased gradually and dramatically with age, reaching a 1.25-fold change by old age. With aging, the differences became evident in the middle-age ($274.9 \pm 57.0$ mg/dL vs. $317.5 \pm 56.9$ mg/dL; $p < 0.05$), and were even more pronounced in the old population ($274.9 \pm 57.0$ mg/dL vs. $344.3 \pm 58.3$ mg/dL; $p < 0.001$) (Fig. 1B). As expected, age was positively associated with Fbg levels ($r = 0.445$, $p < 0.0001$) (Fig. 2B).

### Effect of ABO blood type, age and gender on VWF levels

As shown in Figs. 1C–1E, plasma levels of VWF (VWF:Ag, VWF:CBA and VWF:Rcof) were significantly higher ($110.3 \pm 29.2\%$ vs. $88.1 \pm 24.6\%$, $120.7 \pm 30.1\%$ vs. $98.4 \pm 27.2\%$, and $132.1 \pm 30.7\%$ vs. $108.4 \pm 28.4\%$, respectively; all $p < 0.05$) in non-O than in O group. VWF:Ag showed a gradual increase with age, reaching a 1.47-fold increase by old age ($87.3 \pm 19.4\%$ vs. $116.1 \pm 25.7\%$ vs. $128.6 \pm 30.3\%$, $p < 0.01$), similar to FVIII:C, as the antigen was strongly associated with FVIII:C levels ($r = 0.746$, $p < 0.0001$) (Fig. 3A). Similarly, VWF:CBA and VWF:Rcof were also positively associated with age. In the three age categories, VWF:CBA levels were $79.3\% \pm 17.4\%$, $95.5 \pm 24.8\%$ and $110.6 \pm 32.3\%$, respectively. And the values for VWF:Rcof were $96.5\% \pm 15.4\%$, $120.2 \pm 22.8\%$ and $126.6 \pm 27.3\%$, respectively. And, more remarkable, there was definitely correlation among age with VWF:Ag, VWF:CBA and VWF:Rcof ($r = 0.410$, $r = 0.401$, and $r = 0.589$, resp.; $p < 0.001$) (Figs. 2C–2E). Like VWF:Ag, VWF:CBA and VWF:Rcof were also positively correlated with FVIII:C ($r = 0.746$ and $r = 0.576$, $p < 0.0001$) (Figs. 3B and 3C). Furthermore, VWF:Ag was highly associated with VWF:CBA and VWF:Rcof ($r = 0.847$ and $r = 722$, $p < 0.0001$)

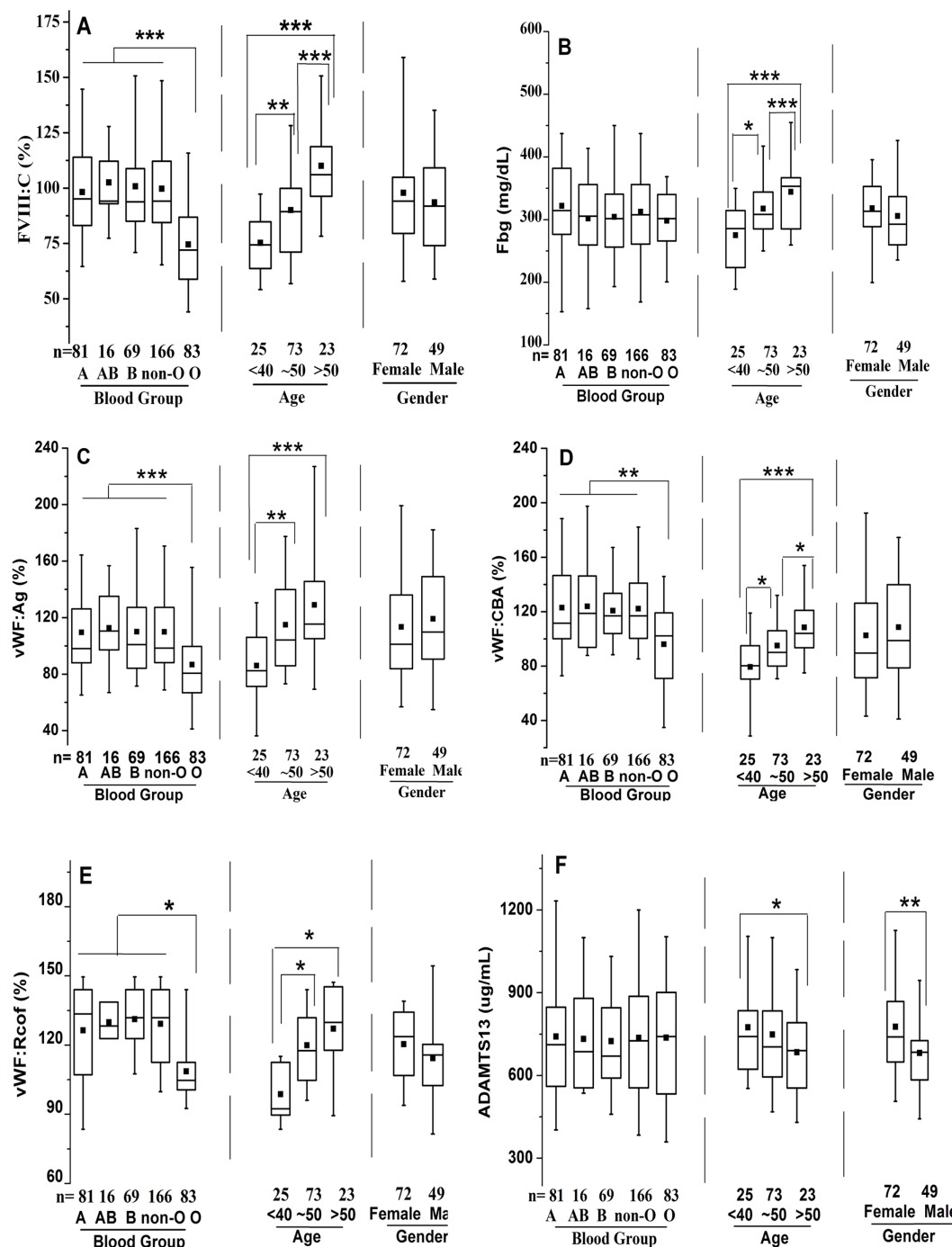

**Figure 1** **Effects of ABO type, age, and gender on distribution of plasma FVIII:C, Fbg, VWF:Ag, VWF:CBA, VWF:Rcof, and ADAMTS13 antigen levels.** The box plots encompass the 25th–75th quartiles, with the center lines (—) representing the median values and the solid boxes (■) showing the mean values. The whisker plots represent 95% CI. *$p < 0.05$; **$p < 0.01$; ***$p < 0.001$. The effects of ABO type and age were calculated using one-way ANOVA followed by LSD post hoc test, whereas the influence of gender was calculated using two-tailed unpaired Student's $t$-tests. Non-O group is composed of A, B and AB groups. (A) FVIII:C; (B) Fbg; (C) VWF:Ag; (D) VWF:CBA; (E) VWF:Rcof; (F) ADAMTS13 antigen.

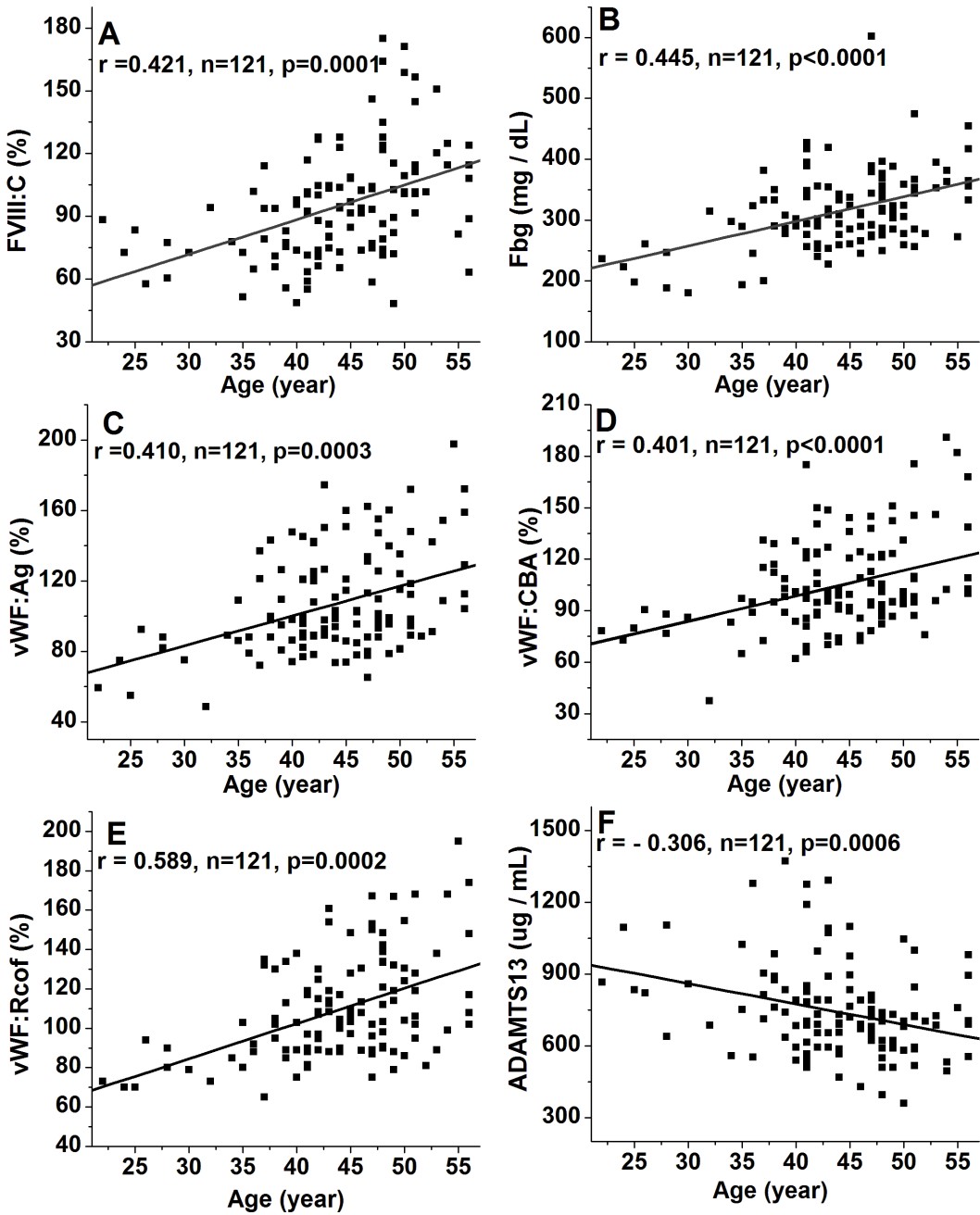

**Figure 2  Relationship between age and plasma levels of FVIII:C, Fbg, VWF(Ag, CBA, and Rcof), and ADAMTS13 antigen.** The diagonal lines indicate linear regression, and Poisson analysis was used. (A) Relationship between age and FVIII:C; (B) Relationship between age and Fbg; (C) Relationship between age and VWF:Ag; (D) Relationship between age and VWF:CBA; (E) Relationship between age and VWF:Rcof; (F) Relationship between age and ADAMTS13 antigen. Strong positive linear correlations are present for FVIII:C, Fbg, VWF:Ag, VWF:CBA, and VWF:Rcof, whereas a negative association is present for ADAMTS13 antigen.

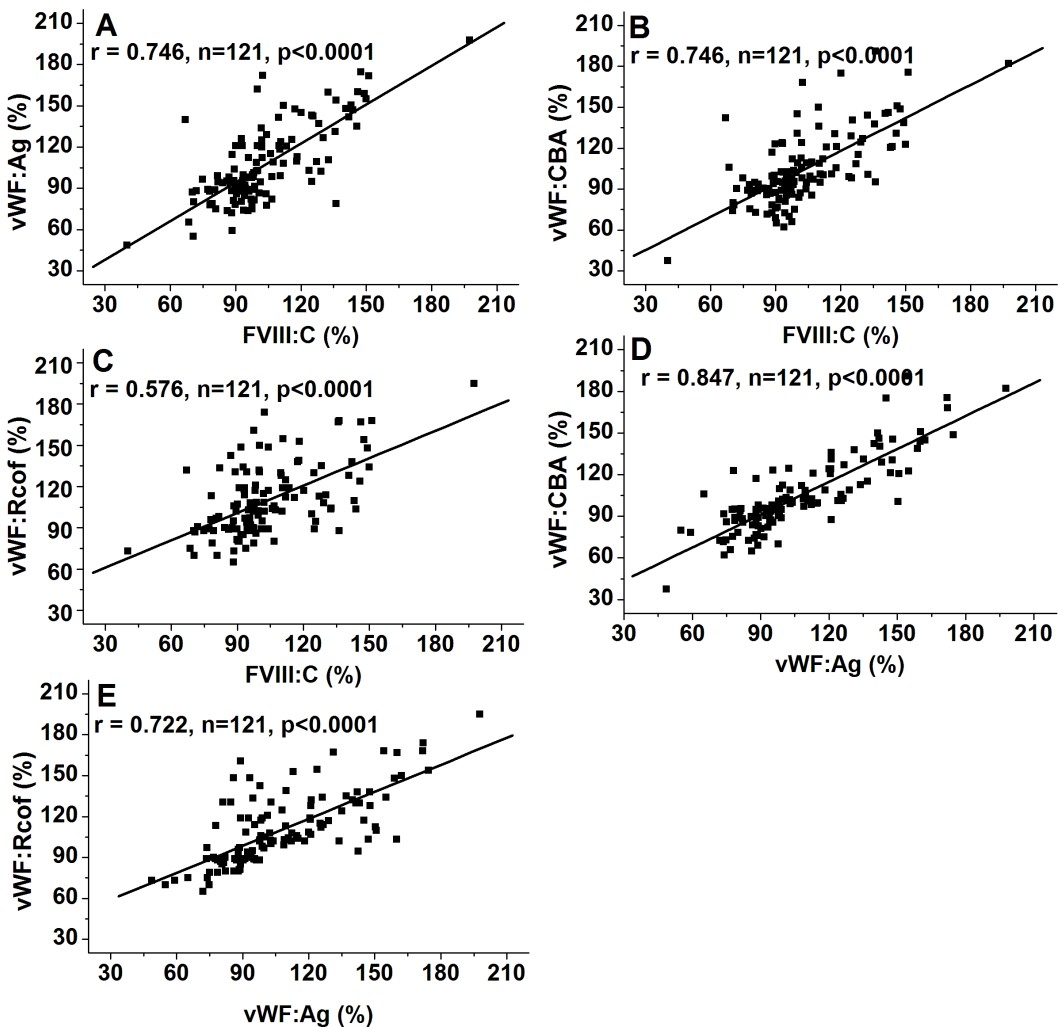

**Figure 3** Associations of FVIII:C and VWF:Ag, VWF:CBA and VWF:Ag, and VWF:Rcof and VWF:Ag. Bivariate correlation analysis was used. (A) Association of FVIII:C and VWF:Ag; (B) Association of VWF:CBA and FVIII:C; (C) Association of VWF:Rcof and FVIII:C; (D) Association of VWF:CBA and VWF:Ag; (E) Association of VWF:Rcof and VWF:Ag. VWF:Ag visibly correlated with FVIII:C, VWF:CBA, and VWF:Rcof.

(Figs. 3D and 3E). Nevertheless, no significant difference in VWF levels was observed between male and female (Figs. 1C–1E).

The corresponding ratios are summarized in Table 2. ABO blood group, age, and gender showed no effect on the ratios of VWF:Ag/FVIII:C, VWF:Ag/VWF:CBA, and VWF:Ag/VWF:Rcof (all $p > 0.05$). However, VWF:Ag-to-ADAMTS13 antigen ratio evidently increased with age (0.91 for <40 and 1.33 for >50; $p < 0.05$) and was markedly lower in females than in males (1.01 vs. 1.24; $p < 0.05$).

## Effect of ABO blood type, age and gender on ADAMTS13 antigen

Clearly different from the other plasma proteins we assessed in this study, ABO blood group showed no effect on plasma ADAMTS13 antigen level. ADAMTS13 antigen levels did

**Table 2 Influences of ABO blood group, age, and gender on corresponding ratios.**

| Corresponding ratios | Blood group | | Age (year) | | | Gender | |
|---|---|---|---|---|---|---|---|
| | Non-O | O | <40 | 40–50 | >50 | Female | Male |
| VWF:Ag/FVIII:C | 1.14 ± 0.44 (1.02–1.25) | 1.21 ± 0.49 (1.10–1.33) | 1.12 ± 0.56 (0.99–1.55) | 1.23 ± 0.49 (1.03–1.31) | 1.17 ± 0.52 (0.79–1.21) | 1.14 ± 0.56 (0.99–1.30) | 1.16 ± 0.62 (0.95–1.38) |
| VWF:Ag/VWF:CBA | 0.96 ± 0.21 (0.87–1.05) | 0.90 ± 0.19 (0.82–0.95) | 1.10 ± 0.20 (0.95–1.23) | 1.21 ± 0.33 (0.92–1.44) | 1.16 ± 0.18 (0.98–1.34) | 1.05 ± 0.20 (0.95–1.16) | 1.08 ± 0.19 (0.99–1.17) |
| VWF:Ag/VWF:Rcof | 0.87 ± 0.46 (0.69–1.58) | 0.82 ± 0.28 (0.60–0.98) | 0.91 ± 0.36 (0.69–1.38) | 0.97 ± 0.21 (0.81–1.22) | 1.01 ± 0.27 (0.85–1.23) | 0.97 ± 0.52 (0.76–1.42) | 1.05 ± 0.43 (0.71–1.13) |
| VWF:Ag/ ADAMTS13[a] | 1.14 ± 0.40 (0.93–1.30) | 0.94 ± 0.45 (0.78–1.10) | 0.91 ± 0.36[b] (0.75–1.08) | 1.10 ± 0.57 (0.93–1.27) | 1.33 ± 0.48[b] (1.12–1.55) | 1.01 ± 0.46[c] (0.89–1.14) | 1.24 ± 0.53[c] (1.05–1.47) |

Notes.

Data are presented as mean ± SD, followed by the range of the results (95% CI) in brackets.

[a]ADAMTS13 antigen level was converted to percentage based on the theoretical value (i.e., 100% in 1 mL of reference plasma).

[b]The ratio of VWF:Ag/ADAMTS13 of the <40 y group was significantly lower than that of the >50 y group. $P = 0.016$, one-way ANOVA followed by LSD post hoc test.

[c]The VWF:Ag/ADAMTS13 ratio was lower in females than in males. $P = 0.042$, two-tailed unpaired Student's $t$-test.

not increase significantly by middle-age (676.8 ± 174.3 ug/mL vs. 750.7 ± 176.9 ug/mL; $p > 0.05$), however, with aging, they were significantly elevated in the old population (676.8 ± 174.3 ug/mL vs. 780.7 ± 203.3 ug/mL; $p < 0.05$) (Fig. 1F). Hence, there was a moderate negative correlation between age and ADAMTS13 ($r = -0.306$, $p = 0.0006$) (Fig. 2F). Interestingly, there was a clear distinction between genders, where ADAMTS13 antigen levels were observably higher in female than in male (680.8 ± 169.3 ug/mL vs. 772.6 ± 171.9 ug/mL; $p < 0.01$) (Fig. 1F).

## DISCUSSION

We evaluated the influences of ABO blood group, age, and gender on plasma levels of FVIII:C, Fbg, VWF (VWF:Ag, VWF:CBA, and VWF:Rcof), and ADAMTS13 antigen in a Chinese population aged 22 y–56 y. The effects of ABO and age on FVIII and VWF levels have been extensively studied, but this is the first integrated report in a Chinese population. In the present study, ABO blood type had a clear influence on FVIII:C and VWF plasma levels. FVIII and VWF levels were significantly lower in O than in non-O groups, whereas no significant differences were calculated among the non-O groups (A, B and AB). Similar results have been confirmed in other populations in USA (*Miller et al., 2003*), Australia (*Favaloro et al., 2005*), Norway (*Orstavik et al., 1985*), Japan (*Kokame et al., 2011*), Brazil (*Rios et al., 2012*), Italy (*Gallinaro et al., 2008*), Canada (*Albánez et al., 2016*), and so on, which imply that the diversities of FVIII and VWF levels were partially determined by ABO blood types. This result may be explained by the variable clearance of VWF as an effect of the ABO glycosylation patterns. Although the mechanism of the effect of ABO on VWF levels is still unclear, the widely accepted hypothesis is that the significantly lower VWF levels in O-group subjects are due to a shorter VWF survival, which is mainly attributable to a faster clearance (*Albánez et al., 2016*; *Casari et al., 2013*; *Gallinaro et al., 2008*; *Lenting, Christophe & Denis, 2015*; *Song et al., 2015*). In a genome-wide association study, it was found that common variants in *ABO* and *VWF* genes explained about 18.7% of the variations in VWF levels (*Desch et al., 2013*). The analysis of genotype uncovered

that the A1 and B alleles were positively and significantly associated with VWF:Ag, whereas O and A2 alleles were negatively associated (*Desch et al., 2013*; *Song et al., 2015*). Using a phage display system, Desch and colleagues (*2015*) revealed mutations in *VWF* gene (at least in VWF73, D1596–R1668) also resulted in reduced susceptibility to ADAMTS13 cleavage, which challenged the hypothesis that specific ABO glycosylation patterns affect proteolysis of VWF (*Deforche et al., 2015*). ABO-related variations in FVIII levels are primarily mediated through VWF, because VWF is a carrier of FVIII, which protects FVIII from clearance. Surprisingly, *Coppola et al. (2003)* did not observe significant differences in VWF levels between group O and non-O in Italian healthy centenarians, although they still found a clear distinction in younger controls. ABO blood group showed no conspicuous effect on Fbg and ADAMTS13 levels, which is consistent with previous reports (*Albánez et al., 2014*; *Alpoim et al., 2011*), although none of the studies evaluated the relationship of Fbg and ABO type. By contrast, *Mannucci, Capoferri & Canciani (2004)* showed that ADAMTS13 levels are lower in individuals with non-O than O blood type, whereas *Rios et al. (2012)* found that ADAMTS13 is significantly higher in non-O subjects.

Plasma levels of FVIII:C, Fbg, VWF:Ag, VWF:CBA, and VWF:Rcof significantly increased with age (Figs. 1 and 2), as supported by other studies (*Albánez et al., 2016*; *Favaloro et al., 2005*). These may explain the higher incidence of thrombosis in older individuals, since Fbg, FVIII and VWF are the three acknowledged risk factors for thrombosis. An interesting finding is that the effect of age on plasma VWF levels appears to be greater than that of ABO type. As shown in Fig. 1C, mean VWF:Ag is approximately 20.1% (110.3 ± 29.2% vs. 88.1 ± 24.6%, $p < 0.001$) lower in O than in non-O carriers and approximately 32.1% (87.3 ± 19.4% vs. 128.6 ± 30.3%, $p < 0.001$) lower in individuals <40 y than >50 y ($p < 0.0001$). With VWF:CBA and VWF:Rcof, the situation is similar. To the best of our knowledge, we are unaware of any similar published reports. However, *Kadir et al. (1999)* showed that VWF:Ag increased by an average of 0.17 (equivalent to 17%) for each 10-year increase in age, which were comparable with our results. In addition, a decreasing trend in ADAMTS13 antigen level was observed in the inclusive ages (22 y to 56 y) in our study (Fig. 1F), which is supported by the report of Kokame and colleagues (*2011*). By contrast, *Mannucci et al. (2001)* showed that ADAMTS13 is significantly lower ($p = 0.03$) in subjects <65 y than >65, which is comparable with the results reported by *Feys et al. (2007)*. These discrepancies may be the results of different races, numbers of samples, and test methods (antigen or activity) used.

Gender showed no significant effect on the tested parameters (Fig. 1), except the ADAMTS13 antigen. ADAMTS13 level is higher in females than in males in a Japanese population (*Kokame et al., 2011*), which is similar to our results. Consistent with our results, gender-related differences in FVIII (*Favaloro et al., 2005*; *Kadir et al., 1999*; *Madla et al., 2012*), Fbg (*Madla et al., 2012*), and VWF (*Favaloro et al., 2005*; *Kadir et al., 1999*) were not observed. However, *Conlan et al. (1993)* reported that FVIII and VWF levels are significantly higher in females than in males.

We also demonstrated that FVIII:C was strongly correlated with VWF:Ag ($r = 0.746$; $p < 0.0001$), VWF:CBA ($r = 0.746$; $p < 0.0001$), and VWF:Rcof ($r = 0.576$; $p < 0.0001$; Figs. 3A–3C), respectively. Furthermore, there was also a definitely correlation among plasma

VWF:Ag with VWF:CBA ($r = 0.847$; $p < 0.0001$) and VWF:Rcof ($r = 0.722$; $p < 0.0001$; Figs. 3D and 3E). Similarly, *Conlan et al. (1993)* suggested that FVIII and VWF are strongly correlated with each other. *Miller et al. (2003)* showed that VWF parameters (VWF:Ag, VWF:CBA, and VWF:Rcof) are significantly correlated with FVIII and with each other.

In this study, ABO type, age, and gender showed no effect on the ratios of the tested parameters (Table 2), except for VWF:Ag/ADAMTS13. VWF:Ag/ADAMTS13 increased with age, especially between <40 y and >50 y (0.91 vs. 1.33; $p = 0.016$), and was significantly less in females than in males (1.01 vs. 1.24; $p = 0.042$). An explanation for the increase of VWF:Ag/ADAMTS13 with age is that VWF:Ag levels increase with age but ADAMTS13 decreases. In addition, the interpretation for the effect of gender on VWF:Ag/ADAMTS13 is that ADAMTS13 levels are higher in females than in males but no gender-related difference in VWF levels is found in our study. Ratios of VWF antigen and functional activity levels, with theoretical values slightly $\geq 1$, could facilitate our understanding of whether or not the protease is fully enzymatical active. In our study, VWF:Ag/VWF:CBA and VWF:Ag/VWF:Rcof were all quite close to 1 under different conditions (Table 2).

## CONCLUSION

Our study clearly suggested that ABO blood type significantly influenced plasma levels of FVIII:C and VWF (VWF:Ag, VWF:CBA, and VWF:Rcof), but not Fbg and ADAMTS13. All parameters observably increased with age, except ADAMTS which showed a decreasing trend. ADAMTS level was evidently higher in females than in males, whereas no significant differences were observed for FVIII, Fbg, and VWF levels by gender. Moreover, VWF:Ag was strongly correlated with FVIII:C, VWF:CBA, and VWF:Rcof. ABO blood group, age, and gender showed no effect on the corresponding ratios, except on the VWF:Ag-to-ADAMTS13 ratio. In this study, we comprehensively evaluated the influences of ABO blood group, age, and gender on plasma levels of FVIII, Fbg, VWF, and ADAMTS13 in a Chinese population aged from 22 y to 56 y. The diversity of ethnicity in biological activities has been increasingly recognized for physiological processes and disease development, so our results, which provide new information on a Chinese population, are scientifically important.

## ACKNOWLEDGEMENTS

The authors sincerely thank the researchers of Guanghan Plasmapheresis Center and Changning Plasmapheresis Center for recruiting volunteers and detecting the ABO blood group phenotype.

### Funding

This work was supported by the Provincial Science and Technology Support Program of Sichuan (No. 2014SZ0123). The funders had no role in study design, data collection and analysis, decision to publish, or preparation of the manuscript.

## Grant Disclosures

The following grant information was disclosed by the authors:

Provincial Science and Technology Support Program of Sichuan: 2014SZ0123.

## Competing Interests

The authors declare there are no competing interests.

## Author Contributions

- Zongkui Wang conceived and designed the experiments, performed the experiments, analyzed the data, contributed reagents/materials/analysis tools, wrote the paper, prepared figures and/or tables, reviewed drafts of the paper.
- Miaomiao Dou performed the experiments, prepared figures and/or tables.
- Xi Du performed the experiments, contributed reagents/materials/analysis tools.
- Li Ma conceived and designed the experiments, reviewed drafts of the paper.
- Pan Sun performed the experiments, reviewed drafts of the paper.
- Haijun Cao conceived and designed the experiments, contributed reagents/materials/-analysis tools, reviewed drafts of the paper.
- Shengliang Ye performed the experiments, analyzed the data, contributed reagents/materials/analysis tools.
- Peng Jiang performed the experiments, analyzed the data, contributed reagents/materials/analysis tools, prepared figures and/or tables, reviewed drafts of the paper.
- Fengjuan Liu performed the experiments, contributed reagents/materials/analysis tools, reviewed drafts of the paper.
- Fangzhao Lin conceived and designed the experiments.
- Rong Zhang conceived and designed the experiments, contributed reagents/materials/analysis tools, wrote the paper, prepared figures and/or tables, reviewed drafts of the paper.
- Changqing Li conceived and designed the experiments, contributed reagents/materials/analysis tools, wrote the paper, reviewed drafts of the paper.

## Human Ethics

The following information was supplied relating to ethical approvals (i.e., approving body and any reference numbers):

The study was approved by the Department of Public Health of Sichuan Province and the Academic Board of Institute of Blood Transfusion.

## Data Availability

The raw data has been supplied as Data S1.

## Supplemental Information

Supplemental information for this article can be found online at http://dx.doi.org/10.7717/peerj.3156#supplemental-information.

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
