# Peer review of "Influences of ABO blood group, age and gender on plasma coagulation factor VIII, fibrinogen, von Willebrand factor and ADAMTS13 levels in a Chinese population"

_PeerJ, doi:10.7717/peerj.3156_

## Round 0.1 · original submission · Major Revisions

Two expert reviewers have commented on your manuscript. The merit and novelty of your work lies in the analysis of specifically Chinese donors, which is important because population based research is often biased towards Caucasions. However, both reviewers have found major issues that you need to address in order for PeerJ to publish your work. It is important that you answer by genuine scientific argumentation or perform additional experiments to include data that are required by the reviewers to complement your current dataset.

The fact that a significant number (41) of plasmas was clotted and not available for analysis raises questions. For one, how many freeze-thawing steps have ALL samples of this cohort been through? This is an overlooked problem often resulting in thawing of historic samples that have been through cycles of freezing and thawing. These cycles impact protein content including ADAMTS13 and FVIII. This point should be carefully addressed in a resubmission. Furthermore, donor metadata like age & sex are unavailable for a large number of these samples and one reviewer genuinely remarks that this can bias the interpretation for the complete cohort. This point should also be carefully addressed.

The authors should have the manuscript corrected for English language, preferably by a native English speaker.

The discussion section needs to be condensed significantly and focused to the data and what these mean for blood donor, patients and clinical decision making. Finally, the points raised by this editor do not waive the authors from addressing all other points raised by the reviewers. Only then can a decision on publication be made.

Reviewer 1 ·

Basic reporting

The English requires substantial improvement. The authors structurally use the word 'could' incorrectly (Age and gender have been shown to influence VWF and FVIII levels, not 'could influence').

Both the introduction and discussion are not very well focused. The aim of the study could be explained more concisely, and the discussion could be reduced by half without compromising on the message of the paper. It is clear why the authors chose to measure VWF/FVIII in their cohort (well known to be influenced by blood group and age, and associated with thrombotic disease), it is understandable why ADAMTS13 was added (as a regulator of VWF function), but the reason to add fibrinogen to the set of parameters is unclear to me.

The raw data file is not fully clear - there are certain numbers in a different color (why), and a couple of numbers that are in different rows, and therefore do not appear to belong to the dataset.

Experimental design

The paper is relatively straightforward. There is abundant data on determinants of VWF/FVIII levels and relations with thrombosis in the literature, but studies specifically in Chinese populations are lacking. The present study only reports levels of selected hemostatic factors in relation with blood group, age, and gender in Chinese subjects. What is lacking are relations with thrombosis (but that would require a completely different set of subjects), and although it seems that results in Caucasians and other ethnic groups and Chinese are comparable, there is no direct comparison made between Chinese and other ethnic groups.

My main concern regards the population. Although 290 blood donors were eligible for the study, a alarmingly high number of plasma samples (41) were clotted during thawing. This could either indicate a structural problem with blood collection (inadequate mixing of blood and anticoagulant) or with the thawing procedure (was this done at 37 degrees???). In addition, of the 249 samples, only 121 samples had information on gender and age, which makes these analyses flawed (are these 121 samples really representative of the 290 original subjects....??). Finally, there is no information on the health status of the donors ('no prior history of various diseases according to the plasmapheresis center') is too vague. It would be relevant to have some baseline characteristics and an idea of the questionnaire. Were subjects with recent infection, renal insufficiency, etc excluded?

Validity of the findings

See comments under 2 - I think there are substantial issues with the selection of the cohort (lack of baseline information, lots of samples clotted upon thawing, which casts doubt on general quality of the data). The novelty of the findings is limited, and I do not fully agree with the final conclusion that these results are 'scientifically and clinically important'. How would clinical management alter when knowing these correlations?

·

Basic reporting

1. ISTH standard for von Willebrand factor is "VWF" not "vWF". This is meant to match the gene nomenclature and should be altered in this manuscript.

2. This manuscript contains mild grammatical errors of English. Examples include:
line 94 "could" and line 95 and 104. This is an incorrect use of the word "could".
line 91, Plasma vWF "has", not have.

3. Line 103, "unusually ultra-large vWF". Should be just "ultra-large" or "unusually-large" not both.

4. Line 145-146, "with the development of science..." This sentence is confusing and should be re-worded. Also, if the authors intend that there are many ethnicity specific variations in physiological processes, please provide references.

5. Line 159, "Up to 290.." Please state how many volunteers were actually enrolled.

6. Line 169-170, ", which could be used..." This section of the statement should be deleted or reworded as it is redendent or confusing.

7. Line 170. "They..." should be changed to "The samples"

8. Line 302, ""...but we may firstly provide..." Restate to say "this is the first integrated report in a Chinese population".

9. Line 318, "Rather bizarrely..." This wording should be changed to something lest dramatic.

Experimental design

no comment

Validity of the findings

no comment

Additional comments

This is a generally well-written manuscript describing the variation of von Willebrand factor concentrations (VWF:Ag) and activity (VWF:CBA, VWF:Rcof), factor VIII activity and ADAMTS13 concentrations in a Chinese population. Specific covariates reported on were age, sex and ABO blood group serotypes. The authors report on expected increases in VWF and FVIII measurements with age and ABO that were consistent with previous reports in other ethnicities. The authors do an excellent job reporting the raw data and making appropriate plots so that the reader can see the variation as well as read the statistics. The weakness of this study is the small numbers that make it more difficult to generalize to the larger cheese population. I have just three major points.

1. 41 samples (14%) of samples were discarded due to clotting. This loss of samples increases the risk that the observations reported are biased by the possible exclusion of samples with higher levels of fibrinogen, VWF or FVIII. The authors should comment if they think the sample lose was due to improper handling or if they attempted to redraw samples from these volunteers.

2. The authors do an excellent job at referencing previous studies. I would urge them to include genome-wide association studies of VWF, ADAMTS13 or FVIII as well especially in their discussion section. (Feel free to exclude this reviewers studies as there are multiple examples beyond my groups work) These studies had much greater power to detect differences in the distribution of VWF, FVIII or ADAMTS13 based on age, sex or gender. These studies also challenge the notion that FVIII has genetic determinants distinct from VWF. They also challenge the idea that specific ABO glycosylation patterns affect ADAMTS13 activity or proteolysis of VWF, at least in the context of a synthetic substrate activity test (VWF73) or the control of VWF levels (as ADAMTS13 was not a signal in multiple VWF GWAS studies).

3. The discussion section should be condensed and avoid repeated reporting of results. I also think the discussion of the magnitude of the environmental factors, age versus ABO (lines 334 - 343) and the discussion of phenotype ratios (lines 3369-387) should be reduced as they do not add significantly to the primary purpose of the manuscript.

---

## Round 0.2 · accepted · Accept

Your paper has been accepted for publication in PeerJ